# Energy-Aware Dynamic 3D Placement of Multi-Drone Sensing Fleet

**DOI:** 10.3390/s21082622

**Published:** 2021-04-08

**Authors:** Yawen Luo, Yuhua Chen

**Affiliations:** Department of Electrical and Computer Engineering, University of Houston, Houston, TX 77204, USA; yuhuachen@uh.edu

**Keywords:** 3D placement, coverage problem, terrains, drone, remote sensing, energy efficiency

## Abstract

Unmanned Aerial Vehicles (UAVs, also known as drones) have become increasingly appealing with various applications and services over the past years. Drone-based remote sensing has shown its unique advantages in collecting ground-truth and real-time data due to their affordable costs and relative ease of operability. This paper presents a 3D placement scheme for multi-drone sensing/monitoring platforms, where a fleet of drones are sent for conducting a mission in a given area. It can range from environmental monitoring of forestry, survivors searching in a disaster zone to exploring remote regions such as deserts and mountains. The proposed drone placing algorithm covers the entire region without dead zones while minimizing the number of cooperating drones deployed. Naturally, drones have limited battery supplies which need to cover mechanical motions, message transmissions and data calculation. Consequently, the drone energy model is explicitly investigated and dynamic adjustments are deployed on drone locations. The proposed drone placement algorithm is 3D landscaping-aware and it takes the line-of-sight into account. The energy model considers inter-communications within drones. The algorithm not only minimizes the overall energy consumption, but also maximizes the whole drone team’s lifetime in situations where no power recharging facilities are available in remote/rural areas. Simulations show the proposed placement scheme has significantly prolonged the lifetime of the drone fleet with the least number of drones deployed under various complex terrains.

## 1. Introduction

As Unmanned Aerial Vehicles (UAVs) are getting more attractions due to their mobility, convenience and low costs, both industry and academia have investigated and deployed drone-related applications and services [1,2,3,4]. UAV-based remote sensing is one of those promising applications [5,6,7]. Drones as camera/sensor platforms offer key advantages over traditional spaceborne and aircraft methods. Compared to satellite observations, the drone-based systems are more capable of having high spatial resolution images. In addition, drone flights can be scheduled whenever the weather and hardware equipment are permitted, as opposed to spaceborne sensing which requires that the satellite orbits back to where the site of interest is covered [8]. As for systematic aerial photography through flights, it requires more resources including aircraft, pilots as well as photographers and has more constraints such as speeds and altitudes.

Due to the above-mentioned capabilities of drone-based sensing systems, drones find many promising applications in crop science, environmental studies, remote sensing and archaeological research [9,10,11].

### 1.1. Related Work

Most researchers focus on planning the flight trajectory of a single drone to serve the sensing/monitoring mission. In [12], Holness et al. let DJI Phantom 2 Vision+ fly linear transects, which were oriented over the area of interest and allowed for the collection of overlapping imagery. They treated overlapping as a necessary requirement for post-processing image stitching to create the large high-resolution mosaic needed for analysis. Authors in [13] applied multispectral, thermal and visible light imaging modules simultaneously for acquiring diagnostic information to serve smart farming purposes. The drone ACSL-PF1 flew at the altitude of 100 m with the speed of 4 m s${}^{-1}$ over the target area. In [14], a drone scheduling problem in which a drone flew on a fixed path was presented. The authors proposed a dynamic algorithm to optimize the speed of the drone to complete tasks within a certain time and without depleting its battery.

The above works produce good results with applications that only require one-time noncontinuous data, but are not suitable with applications that require monitoring the whole area simultaneously for a certain period of time. Studies on coordinated multi-drone surveillance have gained increasing attention over the past years. The multi-drone surveillance mission requires two critical problems to be resolved: drone placement and battery endurance.

The drone placement is a coverage problem. The work in [15,16,17] studied the 3D placement of drone base station. Unlike remote sensing using visual cameras, it applies the electromagnetic (EM) wave for communication. The EM wave can be treated as a type of active sensing while the camera is passive sensing. Alzenad et al. [15] applied line-of-sight and non-line-of-sight communication models to find the best height for drones that maximizes the covered area with a fixed transmitting power. It then formulated the coverage problem to a 2D placement strategy. The authors in [16] applied the same model from [15] to do the placement with extra environmental parameters provided: urban, suburban and dense urban. Reference [17] proposed a simplified placement scheme that utilized equal circles without overlapping to improve coverage efficiency at the cost of some uncovered zones.

These studies, however, only considered the given landscape as a smooth horizontal plane. They typically obtained the optimal altitude first, then formulated the coverage problem into a 2D placement strategy. In real-world applications, especially for visual sensing, each drone’s coverage varies tremendously as topography changes, which complicates the placement.

The battery endurance plays an important role in designing the drone sensing fleet. Naturally, flying drones have limited battery supplies which are supposed to cover physical motions, message transmissions and data calculation. Since the surveillance job demands a drone fleet to hold its position for certain periods, the drone energy model needs to be explicitly investigated and necessary adjustments should be deployed on drone locations. Some existing drone-related papers took energy efficiency into account and tried to minimize the power of each drone. In paper [15], authors tried to find the smallest coverage circle while supporting the same amount of clients, which saved the drone’s transmitting power. Authors in [18,19] managed to find an optimal travel path to improve energy efficiency.

However, these only consider the total energy consumption without evaluating the impact of bottleneck drones on the lifetime of the drone fleet, which is an important factor in the operations of real-world applications. In addition, the fact that drones typically operate in geographically challenging areas is often ignored. Note that in our paper, the lifetime of the drone fleet is defined as the time of the first drone running out of power, which is widely adopted [20,21].

### 1.2. Our Contribution

To address these issues, in this paper, we propose a 3D placement algorithm for the multi-drone sensing fleet, which targets large areas with continuous omnidirectional surveillance requirements and various geographical conditions. The sensing technology ranges from visible light, multispectral, hyperspectral camera to EM transceivers. Potential applications include the assistance of rescuing missions in flooded regions, temporary recovery of telecommunication systems in earthquake areas and environmental monitoring of areas with rough terrains.

In the 3D placement problem, we consider the visual blocking situation for each drone hovering over various terrains. To save the overall energy consumption, we analyze the drone’s energy model and study the impact of mutual distances among the drones. Considering the finite amount of energy sources as well as the fact that each drone has to work in coordination with remaining drones to share information as required by the application, the lifetime of the entire drone fleet is strongly affected by the lifetime of individual drones, especially those in critical locations. In order to balance the entire network’s energy, we design the routing algorithm that chooses the relaying neighbors and selects the data path to reach the destination. In addition, dynamic adjustments are applied to drones’ locations during the mission to prolong the drone fleet’s lifetime.

In our work, we have the following assumptions: The geographical conditions of the investigated area are known with its 3D dimensional contour map given. There are no charging facilities or battery replacements available. Compared to other research, our work contributes in the following ways: (1) We study the placement of a multi-drone sensing problem that requires continuous omnidirectional monitoring. (2) Our model considers topographical factors, including rough terrain situations where mountains, valleys or other obstacles exist. (3) Our 3D placement, routing scheme and real-time location adjustments aim at improving individual energy efficiency while prolonging the lifetime of the whole drone fleet at the same time.

The rest is organized as follows: First, the system architecture is described in Section 2. More specifically, the system model and energy model are elaborated. Next, research methods are illustrated in Section 3, which shows the details of the mesh simplification, sub-section division, drone placement, routing scheme and dynamic adjustments. Then, Section 4 presents simulation results and evaluates the performance. Finally, Section 5 draws the conclusion.

## 2. System Architecture

### 2.1. System Model

For simplicity, the target area *T* is defined as a square with a side length of *a*. The terrains are generated by translating and scaling Gaussian distributions. A set of drones $D=\{D_{1},D_{2},D_{3},...,D_{n}\}$ are responsible for the area of interest. The number *n* is kept as small as possible to serve the monitoring job. The whole surveillance area is supposed to be covered by the drone fleet’s cameras. Based on the requirements of different applications, the camera can be visible light, multispectral, hyperspectral and even be replaced by EM transceivers. Since our focus is on the placing algorithm and the mission duration, we simplify this model with the most common visible light camera HD 1080p. $\theta_{h}$ and $\theta_{v}$ are the camera’s horizontal and vertical field of view (FOV), respectively. Figure 1 illustrates the drone camera’s FOV and shows its coverage during the mission execution. χ (pixels m${}^{-1}$) is the image resolution parameter required by the surveillance task, e.g., the requirements of χ vary greatly from object identification, motion detection to object tracking. The objectives of the placement algorithm include:building a regional coverage model;determining the least number of drones needed for coverage in a given area;finding optimal drones’ 3D-placement locations to save communication energy;proposing a strategy to prolong the entire drone fleet’s lifetime.

### 2.2. Energy Model

The energy consumption of the drone $D_{i}$ during the mission execution process is given by: (1)$$E_{D_{i}}=E_{flight}+E_{cal}+E_{com},$$
where $E_{flight}$, $E_{cal}$ and $E_{com}$ are the energy consumed during the flight, calculation and communication modes, respectively.

#### 2.2.1. *Flight Mode*

Even without any communication cost, drones need to consume power to maintain flying status, which consumes the majority of the battery energy. Drone’s mechanical energy involves two processes, namely, hover and motion. It can be written as: (2)$$E_{flight}=E_{hov}+E_{mov}.$$

The hover energy $E_{hov}$ can be calculated through: (3)$$E_{hov}=P_{hov}\times t_{hov},$$
where $t_{hov}$ is the time duration of the hover status, and $P_{hov}$ is the hover power. According to the model in [18,22], $P_{hov}$ is a function of the drone’s mass *m*, the number of propellers *n* as well as the propeller’s radius *r*: (4)$$P_{hov}=\sqrt{\frac{\left(mg^{3}\right)}{2\pi r^{2}n\rho }},$$
where *g* and ρ are the Earth’s gravitational acceleration and the air density, respectively. As stated in Equation (Equation 4), we want to have a lighter drone with more and larger propellers to achieve low $P_{hov}$. It is a paradox since larger numbers and sizes of propellers will incur larger mass, which goes against the “lightweight” drones. In addition, there is another antinomy: in the real world, manufacturers intend to equip drones with high-capacity batteries for longer flight time; however, the mass of the battery itself has a linear relationship with its charging capacity.

The above constraints limit the drone’s flight duration based on today’s technology. New low-density materials and high-volume batteries are the technical bottlenecks for energy efficiency research in the flight mode, which is in the domain of mechanical and chemical engineering. We apply the latest commercial drone’s parameters as the basis of our following analysis.

As for the motion energy, it can be obtained by: (5)$$E_{mov}=\int P_{mov}\left(t\right)dt,$$
where $P_{mov}$ is the motion power that measures the “moving” process from one location to another and can be modeled as a linear function of the drone speed *v* [18]:   
(6)$$P_{mov}\left(t\right)=\frac{P_{max}-P_{idle}}{v_{max}}v\left(t\right)+P_{idle},$$
where $P_{max}$ and $P_{idle}$ represent the power of drone when moving at the maximum speed $v_{max}$ and holding the hover status, respectively. To some extent, $P_{idle}$ is equal to $P_{hov}$. $P_{max}$ and $v_{max}$ are usually known drone parameters. The speed *v* is time-variant and is subject to: (7)d=∫v(t)dt,
where *d* is the distance between the departure and destination locations.

#### 2.2.2. *Calculation Mode*

The calculation energy $E_{cal}$ can be written as: (8)$$E_{cal}=\int P_{cal}\left(t\right)dt,$$
where $P_{cal}$ is the power consumption of computation. These calculations range from image processing, pattern identification, object tracking to decision making and event reporting. The critical technologies that impact energy efficiency are the calculation algorithms and data structures that are related to specific tasks and applications. In this paper, we do not restrict the type of applications of the drone fleet. However, for simulation purposes, $P_{cal}$ is simplified as a constant and the detail is shown in the Section 4.

#### 2.2.3. *Communication Mode*

The energy spent in message sharing among drones made up another significant portion of battery consumption. For example, in some applications, surveillance area videos should be transmitted to the server where groups of experts and officials can make real-time decisions. It incurs a large amount of data to be sent from drones’ antennas, which makes the batteries drain faster and it might even exceed the energy consumption of the flight mode. Even for cases where there is no need for real-time video transmissions, the critical event reporting messages and nearby drone information sharing packets can still involve a large energy consumption when the communication distance is far. As we do not restrict the type of applications of the drone fleet, we focus on 3D placement schemes that optimize energy spent in the communication mode.

The communication energy is: (9)$$E_{com}=E_{tx}+E_{rx},$$
where $E_{tx}$ and $E_{rx}$ are the energy required for the transmitter and the receiver, respectively. To simplify the energy consumption during the radio communication process, we adopt the model as depicted in [23]. For transmitter or receiver circuitry, the radio transmission dissipates $E_{elec}$ (nJ bit${}^{-1}$). The $E_{amp}$ (nJ bit${}^{-1}$ m${}^{-2}$) is defined for the transmitter amplifier’s consumption. The energy to transfer a message of $l_{packet}$ bits over distance *d* is: (10)$$E_{tx}\left(l_{packet},d\right)=l_{packet}\times E_{elec}+l_{packet}\times E_{amp}\times d^{2}.$$

The energy consumption to receive the packet is: (11)$$E_{rx}\left(l_{packet},d\right)=l_{packet}\times E_{elec}.$$

## 3. Research Methods

We define f(x,y,z)=0 as the terrain function. The location of the drone $D_{i}$ is denoted as $L_{D_{i}}\left(x_{D_{i}},y_{D_{i}},z_{D_{i}}\right)$. $A_{D_{i}}=\left\{P_{i}|\left(x_{P_{i}},y_{P_{i}},z_{P_{i}}\right)\in R^{3}\right\}$ labeled as the covering region of drone $D_{i}$, is the collection of all points where each $P_{i}\left(x_{P_{i}},y_{P_{i}},z_{P_{i}}\right)$ can be detected by drone $D_{i}$. As a result that the sensing camera is HD 1080p (1920p × 1080p), with the task resolution requirement χ (pixels m${}^{-1}$) known, we can obtain the maximum region a drone can supervise by: (12)$$l=\frac{1920}{x},$$
and
(13)$$w=\frac{1080}{y},$$
where *l* and *w* represent the length in the *X* axis (east–west) and the width in the *Y* axis (north–south), respectively. To meet the resolution requirement, $A_{D_{i}}$ is constrained by: (14)$$A_{D_{i}}\subseteq R_{l\times w},$$
where $R_{l\times w}$ represents an l×w rectangular area. The horizontal and vertical FOV $\theta_{h}$, $\theta_{v}$ are constrained by: (15)$$\frac{\tan\frac{\theta_{h}}{2}}{\tan\frac{\theta_{v}}{2}}=\frac{1920}{1080}\approx 1.78.$$

For convenience, we assume that the drone’s horizontal FOV is always along the *X* axis and the vertical FOV is corresponds to the *Y* axis. Considering the light blockage, the coverage problem can be formulated as follows: For any $P_{i}\left(x_{P_{i}},y_{P_{i}},z_{P_{i}}\right)\in f\left(x,y,z\right)=0$, we say $P_{i}$ is under the surveillance of drone $D_{i}$, if 
(16)$$|z_{P_{i}}-z_{D_{i}}|\times \tan\frac{\theta_{h}}{2}\geq \left|x_{P_{i}}-x_{D_{i}}\right|,$$
and
(17)$$|z_{P_{i}}-z_{D_{i}}|\times \tan\frac{\theta_{v}}{2}\geq \left|y_{P_{i}}-y_{D_{i}}\right|,$$
and no other $P_{j}\left(x_{P_{j}},y_{P_{j}},z_{P_{j}}\right)\in f\left(x,y,z\right)=0$ that satisfies Equations (Equation 16) and (Equation 17) as well as: (18)$$0<\frac{x_{P_{j}}-x_{D_{i}}}{x_{P_{i}}-x_{D_{i}}}=\frac{y_{P_{j}}-y_{D_{i}}}{y_{P_{i}}-y_{D_{i}}}=\frac{z_{P_{j}}-z_{D_{i}}}{z_{P_{i}}-z_{D_{i}}}<1.$$

### 3.1. Terrain Simplification

In real-world surveillance tasks, the target region can stretch for several miles. Time and space complexity are critical bottlenecks for the coverage problem and the drone placement scheme. Considering the target area’s size, it is not possible or even necessary to test feasible drone locations point by point. We only need to try the particular coordinates which are iconic “landmarks”. We simplify the terrains as illustrated in Figure 2. Only the space above the simplified vertices is explored. The given topography of the target area *T* is a height map $H=\{h_{i,j}\left(x_{i},y_{j}\right)|\left(x_{i},y_{j}\right)\in R^{2}\}$.

Simplification is performed as follows:transfer height map *H* into a 3D binary volume array $V=\{v_{i,j,k}\left(x_{i},y_{j},z_{k}\right)|\left(x_{i},y_{j},z_{k}\right)\in R^{3}\}$;extract the isosurface of the volumetric array *V* and store the faces and vertices information as FV;apply a mesh decimation algorithm to simplify the surface and obtain faces and vertices as SFV.

The isosurface extraction and patch reduction algorithm are mature [24,25,26] and can be found in many packages or libraries. One example using MATLAB library is shown in Algorithm 1 for other researchers to replicate our works. In addition, some of the notations in the algorithm are used in later discussion.
**Algorithm 1: **Terrain Simplification: Calculate SFV**Require: **$h_{ij}\left(x_{i},y_{j}\right)>0,\forall h_{ij}\in H$**Ensure: **SFV≠Ø1: V⇐02: **for** i=1 **to** *a* **do**3: **for** j=1 **to** *a* **do**4:  **for** 
k=1 **to** $h_{ij}$5:   V(i,j,k)⇐16:  **end for**7: **end for**8: **end for**9: FV⇐isosurface(V)10: SFV⇐reducepatch(FV,ratio)

### 3.2. Area Division

The given area *T* is supposed to be covered by the drone fleet, where each drone will be assigned to a small section $S_{D_{i}}$. We denote the number of sections as *n*. In papers where a flat plane is assumed for the whole area *T*, the division problem is usually simplified. More specifically, every drone will be distributed with an equal rectangular shape. However, due to the sophisticated terrains in our study, the above algorithm does not apply to our system model. As shown in Figure 3, the coverage of a drone can be stretched when placed over the valley (green line) and shrunk when near the mountain (red line). Hence, we propose our division algorithm, which detects the terrain variances and automatically adjusts each subarea’s range so that the drone can cover as large a region as possible, with the least number of drones needed for the mission. The objective of sub-section division is formulated as: (19)min(n),
subject to: (20)$$\overset{n}{\underset{i=1}{⋃}}S_{D_{i}}=T,$$
and
(21)$$S_{D_{i}}\subseteq A_{D_{i}},\forall i\in \left(1,2,\ldots ,n\right).$$

Since the coverage area $S_{D_{i}}$ is a variable and there is no explicit math model to follow, we adopt the greedy algorithm. We perform a search on the remaining uncovered area to find the largest region for each $S_{D_{i}}$, and a location $L_{D_{i}}\left(x_{D_{i}},y_{D_{i}},z_{D_{i}}\right)$ that allows $A_{D_{i}}$ to cover $S_{D_{i}}$. As we analyzed earlier, $A_{D_{i}}$ can be calculated by Equations (Equation 12)–(Equation 18). However, the terrain function is not available and needs to be inferred first. In Algorithm 1, we have simplified faces and vertices stored as SFV. We can fit the vertices $v_{i,j,k}\left(x_{i},y_{j},z_{k}\right)$ to a function, which approximates the terrain function f(x,y,z)=0 [27,28]. More details are shown in Algorithm 2.
**Algorithm 2: **Area Division: Calculate $S_{D_{i}}$**Require: ** T≠Ø**Ensure: **min(n)1: f(x,y,z)=0⇐fit(SFV)2: n⇐03: **while **T≠Ø **do**4:  n⇐n+15:  find 
the 
bottom 
left 
corner 
(x0,y0)
of *T*6:  len⇐l;
wd⇐w7:  **repeat**8:   flag⇐false9:   $S_{D_{n}}⇐Ø$10:   **for**
 $\forall (x_{i},y_{j})\in T$ **do**11:    **if** 
$\left(x0\leq x_{i}\leq x0+len\right)\;$ **and** 
$\left(y0\leq y_{j}\leq y0+wd\right)$ **then**12:    $S_{D_{n}}⇐S_{D_{n}}\cup \left(x_{i},y_{j}\right)$13:    **end if**14:   **end for**15:  **for** 
$\forall \left(x_{i},y_{j}\right)\in S_{D_{n}}$**do**16:    **for** 
$z_{k}=h_{i,j}\left(x_{i},y_{j}\right)\;$**to** 
$z_{max\;}$**do**17:   
$L_{D_{n}}⇐\left(x_{i},y_{j},z_{k}\right)$18:   $A_{D_{n}}⇐Ø$19:   **for** 
∀(x,y,z)∈SFV **do**20:     **if**
(x,y,z) 
covered 
by 
$D_{n}$ **then**21:    $A_{D_{n}}⇐A_{D_{n}}\cup \left(x,y,z\right)$22:     **end if**23:   **end for**24:    find 
max(x), 
min(x), 
max(y), 
min(y) 
$of\;A_{D_{n}}$25:    **if** $S_{D_{n}}\subseteq A_{D_{n}}$ **and** 
(max(x)−min(x))≤l **and** 
(max(y)−min(y))≤w **then**26:    
flag⇐true;
break27:     **end if**28:   **end for**29:    **if** 
flag==true **then**30:    break31:     **end if**32:   **end for**33:   $len⇐len-\delta_{l};$
$wd⇐wd-\delta_{w}$34:   **until** 
flag==true35:   $T⇐T\backslash S_{D_{n}}$36:  **end while**


### 3.3. Placement

In the placement scheme, we aim to save overall energy consumption, which can be written as: (22)$$\min(\sum_{i=1}^{n}E_{D_{i}}).$$

According to Equations (Equation 1)–(Equation 11), we know that $E_{flight}$ depends on drone’s hardware parameters, and $E_{cal}$ is related to the specific application. We focus on 3D placement schemes that optimize the energy spent in the communication mode $E_{com}$. Equation (Equation 10) shows that distance $d_{i,j}$, between a pair of drones $D_{i}$ and $D_{j}$, has large impacts on the energy consumed at the transmitter $E_{tx}$. We further rewrite the target Equation (Equation 22) as: (23)$$\min(\sum_{i,j=1}^{n}d_{i,j}^{2}),$$
subject to: (24)$$d_{i,j}^{2}=shortest\_path\left(L_{D_{i}},L_{D_{j}}\right),$$
where the shortest path algorithm is applied to calculate the distance by considering the fact that to save the transmitting power during communication, drone $D_{k}$ might be assigned to relay the packets sending from $D_{i}$ to $D_{j}$. More specifically, we adopted Dijkstra’s algorithm [29].

In order to decide the drone location $L_{D_{i}}\left(x_{D_{i}},y_{D_{i}},z_{D_{i}}\right)$, we need to find the collection of all feasible drone locations first. We label the desired collection within subarea $S_{D_{i}}$ as $R_{D_{i}}$. The calculation of $L_{D_{i}}$ is implemented in Algorithm 3. The main steps are as follows:search $R_{D_{i}}$ for every sub-area $S_{D_{i}}$ according to Equations (Equation 12)–(Equation 18);initialize each sub-area’s drone location $L_{D_{i}}$ with a random element of $R_{D_{i}}$;for each sub-area $S_{D_{i}}$, find the best location within $R_{D_{i}}$ that minimizes the total distance from drone $D_{i}$ to $D_{j}$ where j∈(1,2,…,n),j≠i;compare the new drone coordinate set with previous ones;if the differences are greater than a predefined threshold, repeat steps 3 and 4. Otherwise, return current drone fleet coordinates.
**Algorithm 3: **Placement: Calculate $L_{D_{i}}$**Require: **$S_{D_{i}}\neq Ø,\forall i\in \left(1,2,\ldots n\right)$**Ensure: **$min\sum_{D_{i},D_{j}\in D}\left|\right|L_{D_{i}}-L_{D_{j}}{\left|\right|}^{2}$1: **for** 
id=1  **to** *n* **do**2:  **for** 
$\forall \left(x_{i},y_{j}\right)\in S_{D_{id}}$**do**3:    **for** 
$z_{k}=h_{i,j}\left(x_{i},y_{j}\right)$ **to** 
$z_{max}$ **do**4:    $L_{D_{id}}⇐\left(x_{i},y_{j},z_{k}\right)$5:    $A_{D_{id}}⇐Ø$6:    **for** 
∀(x,y,z)∈SFV**do**7:     **if** 
(x,y,z) covered by 
$D_{id\;}$**then**8:      
$A_{D_{id}}⇐A_{D_{id}}\cup \left(x,y,z\right)$9:       **end if**10:   **end for**11:    find 
max(x), 
min(x), 
max(y), 
min(y) 
$ofA_{D_{id\;}}$12:    **if** 
$S_{D_{id}}\subseteq A_{D_{n}}$ **and** 
(max(x)−min(x))≤l  **and** 
(max(y)−min(y))≤w **then**13:    **if** $R_{D_{id}}⇐R_{D_{id}}\cup \left(x_{i},y_{j},z_{k}\right)$14:    
**end if**15:   
**end for**16: **end for**17: **end for**18: **for** 
id=1 **to** *n* **do**19:  $L_{D_{id}}⇐R_{D_{id}}\left(1\right)$20: **end for**21: OldL⇐L22: Dif⇐Threshold+123: **while** 
Dif≥Threshold **do**24:   **for** 
i=1 *n* **do**25:    sum⇐MAX26:    **for** 
$\forall \left(x,y,z\right)\in R_{D_{i}}$ **do**27:     $temp⇐\sum_{j\in (1,2,\ldots ,n)\mathrm{and}j\neq i}Dijkstra\left(L_{D_{j}},\left(x,y,z\right)\right)$28:    **if** 
temp<sum **then**29:    sum⇐temp30:    $L_{D_{i}}⇐\left(x,y,z\right)$31:    **end if**32:   **end for**33:  **end for**34:  Dif⇐||OldL−L||35:  OldL⇐L36: **end while**

### 3.4. Dynamic Adjustment

According to Equation (Equation 10), the radio transmission power significantly increases with distances. When communication packets are delivered between two remote drones, we prefer to use one or more intermediary drones for relaying. As all the drone’s locations are fixed and known, we use Dijkstra’s algorithm to find the shortest path from each drone $D_{i}$ to $D_{j}$. We denote *Q* as the routing table. $Q_{i,j}$ is the next-hop drone when the packet source is $D_{i}$ and the destination is $D_{j}$. Detailed routing table calculation is illustrated in Algorithm 4.
**Algorithm 4: **Routing Table: Calculate $Q_{i,j}$**Require: **$L_{D_{i}}\neq Ø,\forall i\in N=\left(1,2,..,n\right)$**Ensure: **$Q_{i,j}\neq Ø,\forall i,j\in N$1: **for** 
i=1 *n* **do**2:  **for** 
j=1 **do** *n* **do**3:   
$Q_{i,j}=j$4:   $dis_{i,j}=\left|\right|L_{D_{i}},L_{D_{j}}{\left|\right|}^{2}$5:  **end for**6: **end for**7: **for** 
i=1 **to** *n* **do**8:  d=dis9:  M⇐N\(i)10:  temp⇐i11:  **while** 
M≠Ø **do**12:   **for** 
∀j∈M **do**13:    **if** 
$d_{i,temp}+d_{temp,j}<d_{i,j}$ **then**14:     $d_{i,j}=d_{i,temp}+d_{temp,j}$15:     $Q_{i,j}=temp$16:    **end if**17:  **end for**18:  find *j* that $d_{i,j}=min_{k\in M}d_{i,k}$19:  
M⇐M\j20:  temp⇐j21:   **end while**22: **end for**

The initial 3D placement ensures optimal overall energy saving. During the mission execution, the drone at the team center is more likely to relay the most packets and have the least energy left. We further propose a dynamic location switching scheme to prolong the drone fleet’s lifetime. The main idea is to switch the location of the lowest remaining battery drone with the highest remaining battery drone when a threshold condition is met.

Compared to the initial 3D placement, which is performed before the mission with no constraints in time, memory and energy, the dynamic adjustments are calculated by the drones in real time. Each drone maintains a battery volume table, which stores every drone’s remaining energy and will be updated periodically by receiving other drone’s broadcast packets. This somehow implies extra energy consumption. However, comparing to other messages and information exchanged among the drones, the remaining battery information can be sent with a single short packet and it has negligible overheads comparing with the potential gains in extending the lifetime of the fleet. In addition, the maintenance costs of a one-dimensional array are negligible compared to the drone’s other processing tasks. Considering energy efficiency, only the drone with the highest remaining battery will run the dynamic switching algorithm. Denote the highest energy drone as $D_{i}$. By checking its battery volume table, $D_{i}$ will start to execute the switching strategy as follows. First, $D_{i}$ calculates the approximate power of each drone through: (25)$$POWER=\frac{ENERGY_{pre}-ENERGY}{INTERVAL},$$
where $ENERGY_{pre}$ is the old battery volume table and ENERGY is the newly updated table. INTERVAL is the table update period. Next, $D_{i}$ computes the ideal maximum lifetime left $T_{left}$ for the drone fleet via: (26)$$T_{left}=\frac{\sum_{i}ENERGY_{i}}{\sum_{i}POWER_{i}}.$$

$T_{left}$ is the upper limit, which can only be achieved when the entire drone fleet is balanced and every drone has the same power. Then, $D_{i}$ checks whether the lowest battery drone (denoted as $D_{j}$) needs switching according to: (27)$$\frac{ENERGY_{j}}{POWER_{i}}\leq T_{left},$$
where the inequality means the battery capacity of $D_{j}$ is extremely low and a switching process is urgently needed. If the condition in Equation (Equation 27) is met, $D_{i}$ will further compare the movement energy consumption $E_{mov}$ from $L_{D_{i}}$ to $L_{D_{j}}$ and $E_{sw\_ovhd}$ with the switching benefits. $E_{sw\_ovhd}$ is the energy consumption caused by the switching overheads and is calculated by: (28)$$E_{sw\_ovhd}=P_{sw\_ovhd}\times t_{sw},$$
where $P_{sw\_ovhd}$ accounts for the extra power needed for switching and $t_{sw}$ is the time needed for the switching process between $L_{D_{i}}$ and $L_{D_{j}}$. As a result that our works do not restrict the type of applications running on the drone fleet or the specific hardware configurations, and the switching overheads are application/machine-dependent, we simplify $P_{sw\_ovhd}$ as a constant, which is given in Section 4 and identical to $P_{cal}$. Finally, as long as the $E_{mov}$ and $E_{sw\_ovhd}$ are affordable compared to the savings, drone $D_{i}$ will initiate a switching request to $D_{j}$. Once drone $D_{j}$ received the request, it will respond according to its application process. If drone $D_{j}$ is executing a critical program, it will decline the request with a busy response. Drone $D_{i}$ can wait for a certain period before initiating the next request.

To avoid the cases where the high energy drone $D_{i}$ keeps switching with other drones, we design in such a way that once it makes a switch, it will initiate a protocol to inform the next highest energy drone to take over the processing role. $D_{i}$ also broadcasts a message to mark $D_{j}$ as switched, and $D_{j}$ will not be taken into account in future switching processes. Details are shown in Algorithm 5.
**Algorithm 5: **Positions Dynamic Switch: Calculate $T_{life}$**Require: **ENERGY>0**Ensure: **$T_{life}$1: **while** 
min(ENERGY)>0 **do**2: update(ENERGY)3: **if** 
ID==max(ENERGY) **then**4:  $POWER⇐\frac{ENERGY_{pre}-ENERGY}{INTERVAL}$5:  $T_{left}⇐\frac{sum(ENERGY)}{sum(POWER)}$6:  j⇐ lowest battery drone7:  **if** 
$\left(\frac{ENERGY_{j}}{POWER_{ID}}\leq T_{left}\right)$ **then**8:   Calculate SwitchMoveCost(ID,j)9:   Initiate switch protocol10:   **if** request accepted **then**11:    Inform next highest battery drone12:    Broadcast $D_{j}$ has been switched13:  **end if**14:  **end if**15: **end if**16:  **end while**17:  $T_{life}⇐$ Current time

## 4. Results Evaluation

We conducted simulations with rough terrain profiles using MATLAB and the results are analyzed. The proposed scheme is evaluated by comparing the drone team lifetime with/without the 3D-placement algorithm and dynamic location switching strategy. Table 1 defines the parameters used in the simulation.

To ensure that our model applies to various landscapes, we ran three sets of simulations with different terrains as shown in Figure 4: mountain, valley and complex topography.

The first part of the simulations deals with the terrains’ profile simplification. As described in Algorithm 1, the height map of the target area needs to be converted to a volumetric array and be reduced further in scale. Figure 5 shows the terrains’ extraction results. The simplification ratio we applied is 0.0001, which means the number of vertices is approximately reduced by a factor of 10,000. Originally, given the target square area side length as 3000 m, even if we only consider the integer coordinates, there are 9,000,000 possible locations for drone placement in the horizontal plane. After applying the terrain simplification algorithm, there are only 1295, 1196 and 1127 possible locations for mountain, valley and complex topography, respectively. It is clear that the simplified terrain tremendously reduces the number of vertices and edges while preserving critical landmarks and features.

The simulations of the sub-area divisions are immediately followed. The segmentation of various terrains is illustrated in Figure 6. Each sub-region is identified with dots of different colors. For better understanding, each terrain’s division results are demonstrated with both aerial and top-down views. By applying Algorithm 2, we segment the target mountain, valley and complex topography area into 10, 9 and 10 sub-areas, respectively. We can discover that the rectangular coverage over the area with steep slopes is relatively small, which is conformed to the theoretical analysis that the highlands are vulnerable to line-of-sight blockage, resulting in a small surveillance region for the drone. Our area division algorithm can adapt to terrain variances and use the least number of drones for the monitoring mission.

The following simulations deal with the optimal positioning of the drones in order to cover the target area with energy constraints applied. The final 3D-placement locations of the three different terrains are depicted in Figure 7. The black dots represent the drones’ positions over the target zone. A more detailed drone placement coordinates for the three terrains are illustrated in Table 2.

To evaluate our placement algorithm, we further simulated drone task processing and packet transmissions with the energy model applied. Equation (Equation 22) indicates that our placement algorithm aims to save the overall drone fleet energy consumption. In order to show the performance of our 3D positioning algorithm, we made a comparison of drone fleet power between our placement and other random placements, as shown in Figure 8. The reason why random placements are applied is because there is no similar placement strategy available due to the terrain complexity, multi-unit coordination and supervision continuity. To the best of the authors’ knowledge, our work is the pioneer in the topography-aware 3D placement of a multi-drone fleet. The random placements are performed 10 times and only the best performance (minimum power) is used for comparison with our 3D placement algorithm. Clearly, with our 3D-placement scheme applied, the drone fleet consumes less power under all three terrains. This is because our strategy takes drones’ mutual distances into account to minimize the overall communication cost.

Since our proposed 3D placement algorithm is efficient in saving overall drone fleet power, the following simulation results are based on the 3D placement algorithm. To further extend the duration of drone-based tasks, we applied the dynamic position switching algorithm. Figure 9 shows the detailed performance improvements after the switching algorithm is deployed. Since the lifetime parameter in our simulation results can be greatly impacted by the drone’s initial energy $ENERGY_{ini}$, we use the battery utilization *U* instead for evaluation purposes. *U* is obtained by: (29)$$U=\frac{\sum ENERGY_{ini}-\sum ENERGY_{left}}{\sum ENERGY_{ini}},$$
where $ENERGY_{left}$ is the drone’s remaining energy after the sensing mission is terminated. Of course, large battery utilization is preferred, which indicates a longer lifetime. Obviously, our strategy significantly improved battery utilization. Since the battery utilization is proportional to the drone fleet’s lifetime, it is clear that with our dynamic switching scheme applied, the lifetime of mountain topography is increased by 43% (from 55.6 to 79.5). There are 71.3% (from 47.2 to 80.9) improvements for valley topography and complex topography is increased by 59.7% (from 53.1 to 84.8). Due to the unbalanced drone networks, some drones need to relay more data, and some may supervise larger regions and spend more power on data processing. As a result, there are cases when a drone drains all its battery while most of the others have a lot of power left, which causes the entire surveillance mission to be terminated prematurely. Our scheme addresses this problem by dynamically switching low-energy-capacity drones with high-energy-capacity drones.

We also analyzed the communication rate parameter. The data rate of drone’s transmission is proportional to the energy consumption in communication. The terrain applied in this set of simulations is the mountain. Figure 10 illustrates the impact of the transmission rate on the placement strategy. The lifetime prolonging ratio (*LPR*) is defined as: (30)$$LPR=\frac{lifetime_{3D\_DA}-lifetime_{ref}}{lifetime_{ref}},$$
where $lifetime_{3D\_DA}$ is the lifetime with both our 3D placement algorithm and dynamic adjustments algorithm applied. $lifetime_{ref}$ is the lifetime without applying the 3D placement algorithm and dynamic adjustments. It is obvious that our placement scheme is a clear winner, especially as the transmission rate increases.

## 5. Conclusions

In this paper, we studied the dynamic 3D placement of a multi-drone-based sensing system that maximizes the sensing mission time with a minimized number of drones. The target sensing area is of various rough terrains. We analyzed the 3D coverage problem by extracting the terrain features and dividing sub-areas. The drone fleet placement was deployed with energy efficiency taken into account. Moreover, a dynamic position switching algorithm was proposed to prolong the entire drone fleet’s lifetime. Simulations have shown that our placement and routing schemes, as well as the dynamic switching algorithms, are effective in improving the lifetime of the fleet. In the future, a more dynamic positioning algorithm will be studied, where the drone can tour within a specific range, which increases the surveillance area and avoids any potential blind spot.

## Figures and Tables

**Figure 1 sensors-21-02622-f001:**
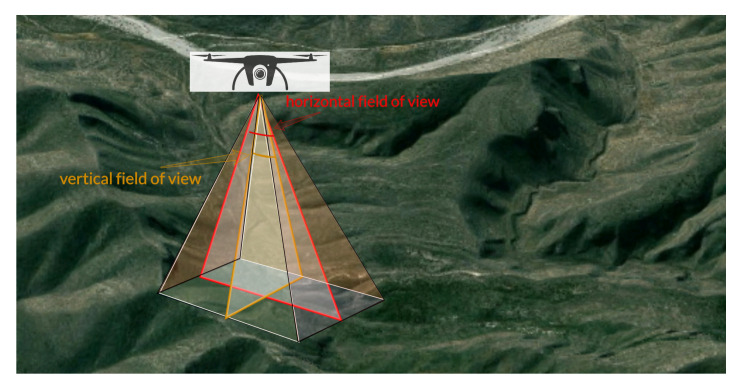
The drone camera’s field of view.

**Figure 2 sensors-21-02622-f002:**
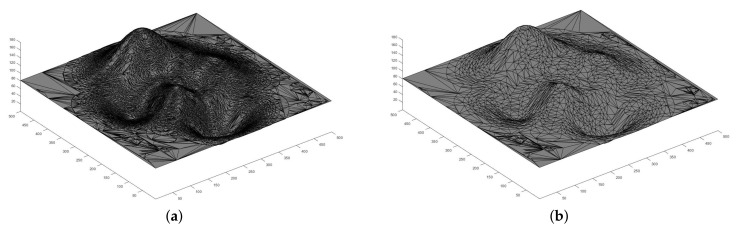
The results of the terrains’ simplification: (**a**) the terrain before simplification, and (**b**) the terrain after simplification.

**Figure 3 sensors-21-02622-f003:**
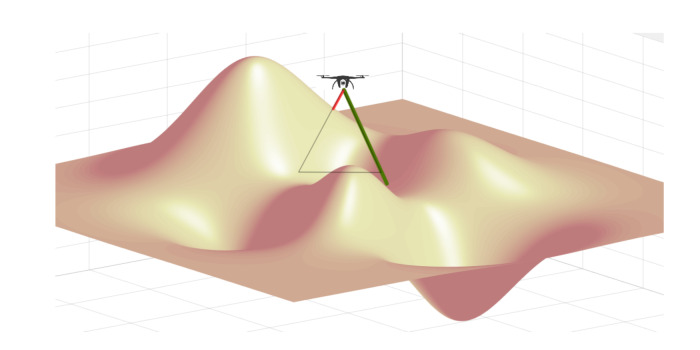
The coverage of the drone is impacted by topography’s variations.

**Figure 4 sensors-21-02622-f004:**
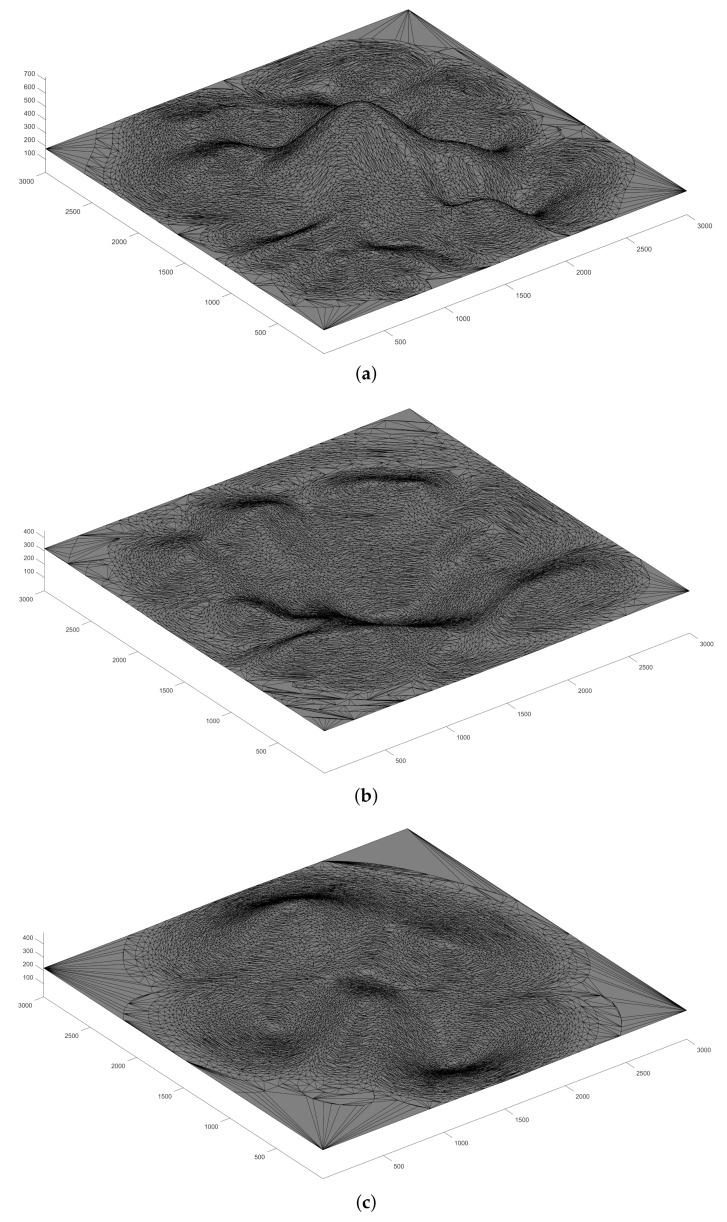
The three different terrains used in our simulations: (**a**) the mountain topography; (**b**) the valley topography and (**c**) the complex topography.

**Figure 5 sensors-21-02622-f005:**
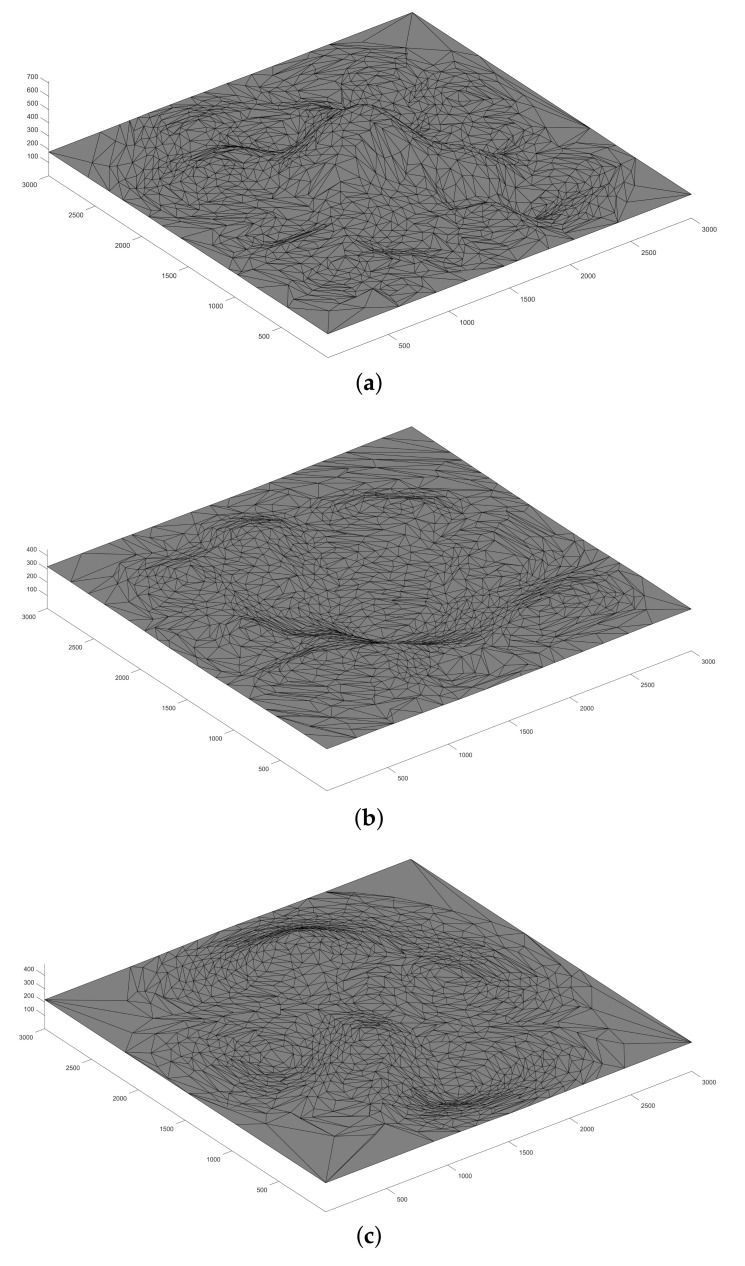
The terrain simplification results of: (**a**) mountain topography; (**b**) valley topography and (**c**) complex topography.

**Figure 6 sensors-21-02622-f006:**
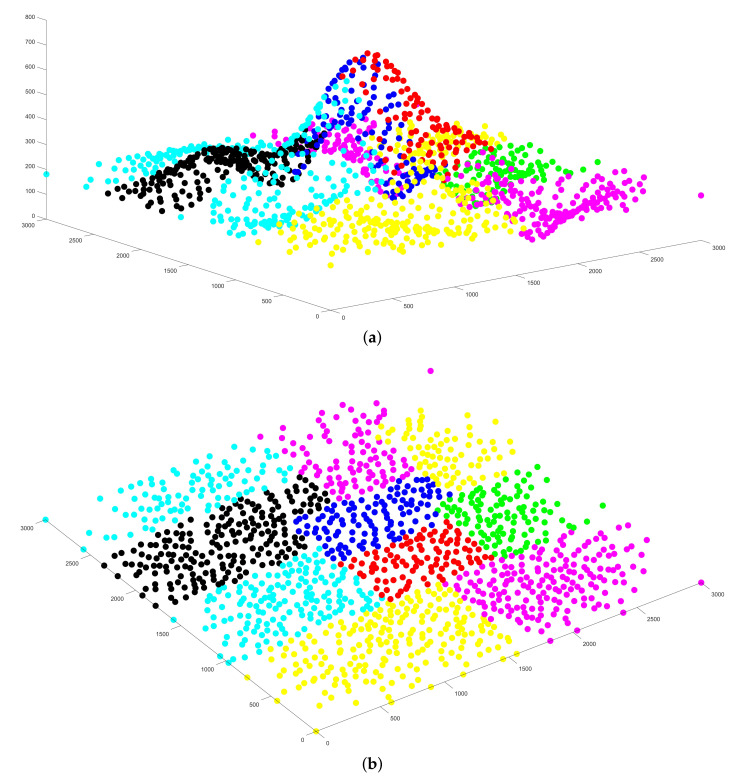
The area division results of: (**a**) mountain topography in aerial view; (**b**) mountain topography in top-down view; (**c**) valley topography in aerial view; (**d**) valley topography in top-down view; (**e**) complex topography in aerial view; and (**f**) complex topography in top-down view.

**Figure 7 sensors-21-02622-f007:**
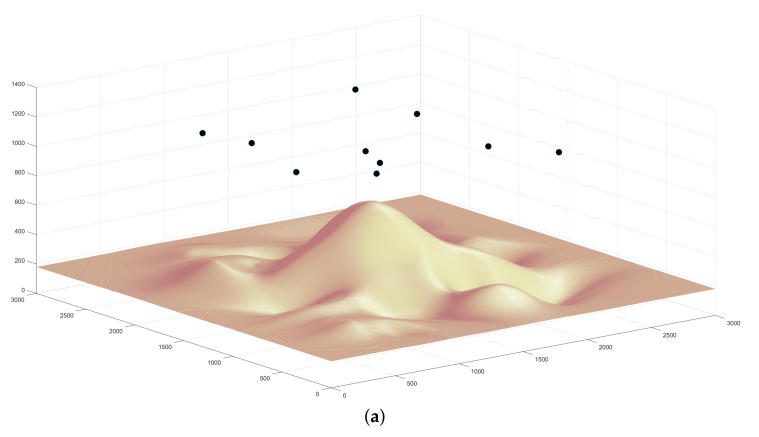
The 3D placement results of: (**a**) mountain topography in aerial view; (**b**) mountain topography in top-down view; (**c**) valley topography in aerial view; (**d**) valley topography in top-down view; (**e**) complex topography in aerial view; and (**f**) complex topography in top-down view.

**Figure 8 sensors-21-02622-f008:**
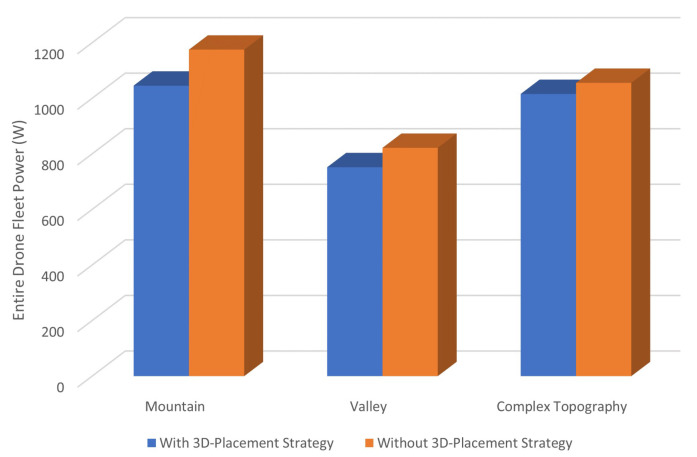
The drone positioning algorithm performance.

**Figure 9 sensors-21-02622-f009:**
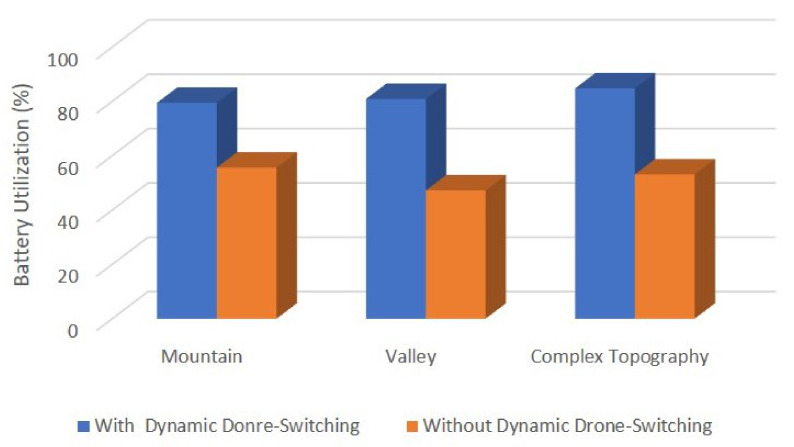
The location dynamic switching algorithm performance.

**Figure 10 sensors-21-02622-f010:**
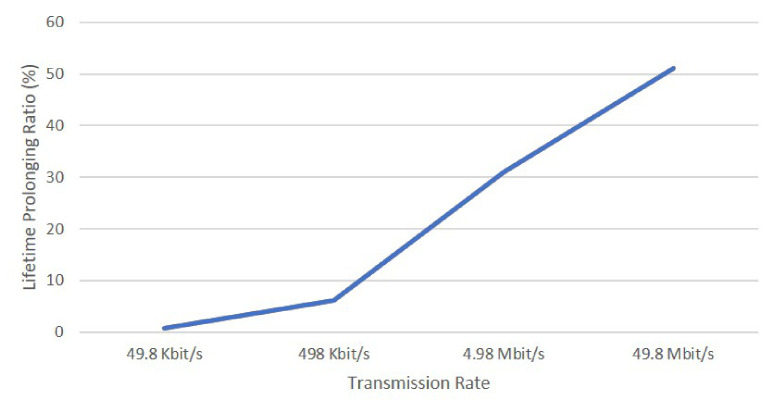
Shows the transmission rate’s impacts on placement strategy.

**Table 1 sensors-21-02622-t001:** The simulation parameters.

Parameters	Values	Parameters	Values
Field side length *a*	3000 m	Drone maximum moving power $P_{max}$	4 W
Horizontal FOV $\theta_{h}$	120${}^{\circ }$	Drone maximum speed $v_{max}$	25 m s${}^{-1}$
Resolution requirements χ	1 pixel m${}^{-1}$	Drone processing power $P_{cal}$	1 W
Drone’s mass *m*	800 g	Transceiver circuitry energy consumption $E_{elec}$	50 nJ bit${}^{-1}$
Number of propellers *n*	4	Transmitter amplifier energy consumption $E_{amp}$	100 pJ bit${}^{-1}$ m${}^{-2}$
Propeller’s radius *r*	12 cm	Drone initial energy storage $ENERGY_{ini}$	34,632 J
Gravitational acceleration *g*	9.8 m s${}^{-2}$	Communication rate within drones	49.8 Mbit s${}^{-1}$
Air density ρ	1.225 kg m${}^{-3}$	Drone switching overheads power $P_{sw\_ovhd}$	1 W

**Table 2 sensors-21-02622-t002:** The drone fleet’s locations of three different topographies.

Drone Location	Mountain	Valley	Complex
$L_{D_{1}}$	(698, 448, 1245)	(778, 425, 1127)	(751, 516, 1074)
$L_{D_{2}}$	(2174, 515, 1134)	(2210, 409, 1200)	(2144, 500, 1132)
$L_{D_{3}}$	(809, 1410, 1031)	(691, 1180, 1125)	(410, 1465, 1085)
$L_{D_{4}}$	(1401, 1330, 1014)	(2166, 1339, 1110)	(1748, 1251, 941)
$L_{D_{5}}$	(2183, 1246, 1016)	(721, 1905, 1018)	(2496, 1308, 964)
$L_{D_{6}}$	(1571, 1697.5, 987)	(1797, 2110, 1045)	(1621, 1926, 1083)
$L_{D_{7}}$	(890, 1969, 1096)	(2553, 2161, 1046)	(2548, 2176, 1052)
$L_{D_{8}}$	(2322, 2153, 1020)	(757, 2703, 1345)	(911, 2405, 1070)
$L_{D_{9}}$	(2168, 2579, 1120)	(2130, 2561, 1229)	(2160, 2505, 1176)
$L_{D_{10}}$	(909, 2494, 1048)	N/A	(885, 2903, 1136)

## Data Availability

Not applicable.

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
