# Peer review of "Energy-Aware Dynamic 3D Placement of Multi-Drone Sensing Fleet"

_sensors, 2021, doi:10.3390/s21082622_

Round 1
Reviewer 1 Report
This paper presents a 3D placement scheme for multi-drone system.
It attempts to deploy drones considering energy model such as energy consumption for flight, calculation and inter-communications.
The proposed model not only
minimizes the overall energy consumption, but also maximizes the whole drone team’s lifetime
There are several issues to be addressed for publication.
1. Energy consumption model is too much simplified ignoring several factors including (but not limited to ) operating system overheads.
2. Simulation results should be expanded.
Authors provides about three sets of results but they are not in-depth simulation to verify contributions of the proposed algorithm.
First of all, authors compared energy efficiency of the proposed platform with "random deployment" policy.
It would be better if they compare the performance with other placement algorithms.
3. In the case of simulation results presented in Figure 9,
it is trivial that drone-switching would provide better "total energy utilization" than a scheme without switching.
The results are not surprising because drone switching obtain the gain at the const of switch overheads.
It would be better to present the overheads of drone switching as well.
Especially, during drone switching there would exist some "non-monitoring " periods because drones are moving to switch their locations and jobs.
4. Minor comments
- There are several sentences (even incomplete sentences) are boldfaced without any specific reason.
(e.g., section 2.1.2.3 on page 5)
5. Some acronyms are not self-contained.
Reviewer 2 Report
In the manuscript, the authors have proposed an energy-aware dynamic 3D placement scheme for the fleet of drones. Several related issues including terrain simplification, area division, placement, and dynamic adjustment are resolved. Simulation works have been conducted to evaluate the performance as well. However, several concerns need to be addressed before the acceptance.
- The authors claim that “few papers have employed multiple drones in a coordinated way to perform surveillance task as a team”. Have the authors considered the UAV swarm or FANET? The coordination of multiple drones is a hot research topic recently.
- Could the authors please reorganize the manuscript? In the current version, it seems that the system architecture, problem statement, proposed methods are all packed into a single section, namely, Section II.
- Could the authors please use a letter or a Greek letter to represent the area in (14) instead of an exact “square”?
- Is it necessary to list Algorithm 1? It seems that the Algorithm is only used to convert height map to volume array and call the algorithms from MATLAB library.
- In Subsection 2.2.4 Dynamic Adjustment, the authors claim that the drone with the highest remaining battery will run the dynamic switching algorithm. Does it assume that all the drones exchanging their remaining battery all the time and every drone holds a copy of the battery volume table? If so, does it imply extra energy consumption?
- In the section of results evaluation, the results are provided with only a few words of description. Could the authors please provide a more insightful analysis of the results?
- What is the definition of the lifetime? Is it the time of the first UAV running of power OR all the UAVS in the fleet running out of power?
Reviewer 3 Report
The paper is well written and the framework is fluent. It presents a 3D placement algorithm based on a multi-drone platform. The algorithm not only improves energy efficiency but also prolongs the lifetime of the whole drone fleet based on consideration of topographical factors. However, there are still some problems to be concerned
1: Please also check the grammar carefully.
2: The coordinate axes of Figure 5, Figure 6 and Figure 7 are not clear.
3: Figures should be captioned correctly. For example, in Figure 8" With 3d-placement Strategy" and " Without 3d-placement Strategy".
4: Some of the results lack data support, such as Figure 8.
5: The format of the References also should be concerned such as Ref.26.
6: One paper related to UAV navigation, the authors may introduce this paper as following.
Song He, Xingwu Chen, Mao-Hsiung Hung, Xiaoyong Chen and Yufeng Ji, “Steering Angle Measurement of UAV Navigation Based on Improved Image Processing”, Journal of Information Hiding and Multimedia Signal Processing, Vol. 10, No. 2, pp. 384-391, March 2019
Round 2
Reviewer 1 Report
Authors have reflected most of comments. Some minor spell checks would be helpful for final publication.
Author Response
Dear Reviewer,
Thank you for your time and effort on our manuscript! Your previous comments and suggestions really help improve our paper. We have carefully checked our spellings and made a further revision. Please refer to our revised manuscript for detailed clarifications.
Reviewer 2 Report
Many thanks for the responses from the authors. Most of my concerns have been addressed. I have no further comments.
Author Response
Dear Reviewer,
Thank you for your time and effort on our manuscript! Your previous comments and suggestions really help improve our paper. And we are glad that our early responses have addressed your concerns properly.